# α-Acylamino-β-lactone *N*-Acylethanolamine-hydrolyzing Acid Amidase Inhibitors Encapsulated in PLGA Nanoparticles: Improvement of the Physical Stability and Protection of Human Cells from Hydrogen Peroxide-Induced Oxidative Stress

**DOI:** 10.3390/antiox11040686

**Published:** 2022-03-31

**Authors:** Agnese Gagliardi, Roberto Molinaro, Massimo Fresta, Andrea Duranti, Donato Cosco

**Affiliations:** 1Department of Health Sciences, University “Magna Græcia” of Catanzaro, Campus Universitario “S. Venuta”, 88100 Catanzaro, Italy; gagliardi@unicz.it (A.G.); fresta@unicz.it (M.F.); 2IRCCS Ospedale San Raffaele srl, 20132 Milan, Italy; molinaro.roberto@hsr.it; 3Department of Biomolecular Sciences, University of Urbino Carlo Bo, Piazza del Rinascimento, 61029 Urbino, Italy

**Keywords:** PLGA nanoparticles, photostability, antioxidant activity, *N*-acylethanolamine-hydrolyzing acid amidase (NAAA), NAAA inhibitors, URB866

## Abstract

*N*-Acylethanolamine acid amidase (NAAA) is an N-terminal cysteine hydrolase that preferentially catalyzes the hydrolysis of endogenous lipid mediators such as palmitoylethanolamide, which has been shown to exhibit neuroprotective and antinociceptive properties by engaging peroxisome proliferator-activated receptor-α. A few potent NAAA inhibitors have been developed, including α-acylamino-β-lactone derivatives, which are very strong and effective, but they have limited chemical and plasmatic stability, compromising their use as systemic agents. In the present study, as an example of a molecule belonging to the chemical class of *N*-(2-oxo-3-oxetanyl)amide NAAA inhibitors, URB866 was entrapped in poly(lactic-co-glycolic acid) nanoparticles in order to increase its physical stability. The data show a monomodal pattern and a significant time- and temperature-dependent stability of the molecule-loaded nanoparticles, which also demonstrated a greater ability to effectively retain the compound. The nanoparticles improved the photostability of URB866 with respect to that of the free molecule and displayed a better antioxidant profile on various cell lines at the molecule concentration of 25 μM. Overall, these results prove that the use of polymeric nanoparticles could be a useful strategy for overcoming the instability of α-acylamino-β-lactone NAAA inhibitors, allowing the maintenance of their characteristics and activity for a longer time.

## 1. Introduction

The fatty acid ethanolamides are a family of endogenous lipid agonists of peroxisome proliferator-activated receptor-α which include palmitoylethanolamide (PEA), a molecule that plays important roles in the control of pain, inflammation and energy balance [1], and in counteracting hepatic oxidative stress [2]. The degradation of PEA or other saturated and monounsaturated fatty acids occurs through *N*-acylethanolamine acid amidase (NAAA) [3,4], an *N*-terminal cysteine nucleophile (Ntn) hydrolase belonging to the choloylglycine hydrolase family and characterized by the ability to cleave nonpeptide amide bonds [5]. NAAA is highly active at acidic pH and only marginally at a pH of around 7 by auto-proteolysis, and it generates a catalytically competent form of the enzyme [6]. It has been proposed as a pharmacological target because NAAA inhibitors administered locally to inflamed tissues where PEA biosynthesis is downregulated are able to restore the physiological levels of the molecule [7]. In particular, this endogenous ligand inhibits peripheral inflammation and mast cell degranulation and has been shown to have strong neuroprotective and antinociceptive effects in rat and mouse models [8,9,10,11]. In this context, the screening of a series of molecules containing chemical functions able to react via nucleophilic linkage with the thiol group of the catalytic cysteine of NAAA, referred to as “cysteine warheads”, is a suitable strategy for inhibiting the enzymatic activity by means of a rapid, noncompetitive and reversible mechanism. Consequent design and biological tests and structure–activity relationship (SAR) studies led to the identification of *N*-(2-oxo-3-oxetanyl)amide as the core of the NAAA inhibitors and showed that their activity was influenced by the β-lactone ring [12,13]. Indeed, several β-lactone derivatives inhibited NAAA, with an IC_50_ in the low nanomolar range, restoring PEA levels after exposure to the pro-inflammatory stimuli in several cell lines. However, these β-lactone derivatives, although showing a crucial inhibitory activity for NAAA, evidenced low chemical and plasma stability, with a half-life of less than 20 min [14]. Significant and consequent measures were taken in order to increase the chemical stability of these compounds, but neither the threonine derivatives, which have a methyl substituent at the β-carbon of the lactone ring [14], the replacement of this group with bulkier electron donor alkyl groups, such as ethyl, isopropyl or *tert*-butyl [15], nor even the introduction of a carbamate group in the place of the amide group, although leading to the discovery of excellent NAAA inhibitors [16,17], allowed for a significant improvement in the situation [14,15]. However, not even attempts to allow for the pharmacokinetic characteristics of these inhibitors through the amelioration of the biological stability have led to promising results [14]. In the scenario of α-acylamino-β-lactone derivatives, the present investigation was carried out on (*S*)-*N*-(2-oxo-3-oxetanyl)-2-naphthamide (URB866), a 2-naphthyl derivative endowed with excellent potency as an NAAA inhibitor (IC_50_ = 160 ± 40 nM) [13], the profile of which is influenced by basic chemical characteristics common to the class (instability due to tension of the lactone ring and the presence of an amide moiety) [14] but also its propensity to undergo oxidative metabolism in the naphthalene aromatic ring. On the other hand, specific in-depth studies on the molecule conducted through high-performance liquid chromatography with UV detection coupled with mass spectrometry analysis confirmed the instability of α-acylamino-β-lactone NAAA inhibitors at neutral and acidic pH [14]. The molecule then became unsuitable for oral administration, but potentially useful for the treatment of localized inflammation by topical application. Based on these findings, a strategy for overcoming the problem of instability is the encapsulation of the compound in drug delivery systems able to preserve its pharmacological efficacy [18,19,20]. Several nanomedicines have been developed in recent decades in order to provide a wide range of advantages, such as protection of the entrapped compound(s), modulation of the solubility, bioavailability and pharmacokinetic profiles of the drugs, a decrease in the side effects and the possibility of obtaining localization in a specific body compartment [21]. In this regard, poly (lactic-co-glycolic acid) (PLGA)-based nanoparticles have been proposed due to the peculiar properties of this material [22,23,24]. Indeed, PLGA is a biodegradable and biocompatible polymer with an ample range of degradation times that may be modified by varying the molecular weight and the chemical composition (lactide/glycolide ratio) [25,26,27]. It has been widely used as a polymer matrix for the entrapment of drugs and various other macromolecules such as DNA, RNA and peptides [28,29,30]. Over 60 PLGA-based drug formulations, particularly microparticle depot preparations, are available on the market, and they include Decapeptyl^®^ (the first PLGA formulation containing triptorelin), Lupron Depot^®^ (leuprolide acetate), Nutropin Depot^®^ (somatropin), Suprecur^®^ MP (buserelin acetate), Sandostatin^®^ LAR Depot (octreotide acetate), Somatuline^®^ LA (lanreotide acetate), Trelstar™ Depot (triptorelin pamoate), Vivitrol^®^ (naltrexone) and Risperdal^®^ Consta™ (risperidone), as well as PLGA-based implants (e.g., Zoladex^®^, Ozurdex^®^, Profact^®^ Depot and Durysta™, containing goserelin acetate, dexamethasone, buserelin and bimatoprost, respectively) and in situ forming implants based on Atrigel^®^ technology, i.e., Eligard^®^ (leuprolide acetate) [21,31,32]. The fact that PLGA-based products are approved for clinical use suggests that PLGA nanomedicine could have a bright future. Moreover, the approval of the PLGA polymer by the US Food and Drug Administration (FDA) and by the European Medicines Agency (EMA) is an additional benefit for the investigation and development of these systems [33]. Based on these findings, the aim of this study was to overcome the rapid degradation of URB866 by means of its entrapment within PLGA nanoparticles. To the best of the authors’ knowledge, this is the first time that a compound of this class has been entrapped in a polymeric system. In detail, physicochemical characterization of PLGA nanoparticles containing URB866 was performed, and the question of whether nanoencapsulation can avoid the degradation of the molecule when it is exposed to ultraviolet (UV) irradiation was evaluated. Moreover, the best formulation was tested in vitro on two human cell lines (C28 and NCTC2544) with the goal of investigating its antioxidant activity with respect to the free compound.

## 2. Materials and Methods

### 2.1. Materials

PLGA (75:25, molecular weight 66–107 kDa), C-28 cells, 3-[4,5-dimethylthiazol-2-yl]-3,5-diphenyltetrazolium bromide salt (used for MTT tests), phosphate-buffered saline (PBS) tablets, dimethyl sulfoxide (DMSO) and amphotericin B solution (250 μg/mL) were purchased from Sigma Aldrich (Milan, Italy). Poloxamer 188 (PLX188) was purchased from BASF (Ludwigshafen, Germany). NCTC2544 cells were provided by the Istituto Zooprofilattico Sperimentale della Lombardia e dell’Emilia Romagna. Minimum essential medium (DMEM) with glutamine, trypsin/ethylene diamine tetraacetic acid (EDTA) (1×) solution and fetal bovine serum were obtained from Gibco (Life Technologies, Monza, Italy). All other materials and solvents used in this study were of analytical grade (Carlo Erba, Milan, Italy).

### 2.2. Synthesis and Characterization of URB866

(*S*)-*N*-(2-Oxo-3-oxetanyl)-2-naphthamide (URB866) was synthesized as described below and in Figure 1. To a stirred mixture of (*S*)-2-oxo-3-oxetanylammonium, toluene-4-sulfonate (1, 0.260 g, 1 mmol) in dry CH_2_Cl_2_ (4 mL), at 0 °C and under N_2_ atmosphere, Et_3_N (0.304 g, 0.42 mL, 3 mmol) and 2-naphthoyl chloride (2, 0.286 mg, 1.5 mmol) were added. The mixture was stirred at 0 °C for 0.5 h and at room temperature for 2 h, and then concentrated. Purification of the residue by column chromatography (cyclohexane/EtOAc 1:1) and recrystallization from MeOH gave URB866 as a white solid.

Melting points were determined on a Büchi B-540 capillary melting point apparatus. (Büchi Labortechnig AG, Flawil, Switzerland). The structure of URB866 was unambiguously assessed by MS, ^1^H NMR, ^13^C NMR and IR. MS (EI) spectra were recorded with a Fisons Trio 1000 (70 eV) spectrometer (Ipswich, United Kingdom). ^1^H NMR and ^13^C NMR spectra were recorded on a Bruker AC 400 or 101 spectrometer (Bruker, Billerica, Massachusetts, USA), respectively, and analyzed using the Top Spin 1.3 software package. IR spectra were obtained on a Nicolet Avatar 360 FT spectrometer (Thermo Fisher, Waltham, Massachusetts, USA). Optical rotations were measured on a Perkin Elmer 241 digital polarimeter (PerkinElmer, Waltham, Massachusetts, USA) using a sodium lamp (589 nm) as the light source. Column chromatography purifications were performed under “flash” conditions using Merck 230–400 mesh silica gel (Darmstadt, Germany). Thin Layer Chromatography was carried out on Merck silica gel 60 F254 plates, which were visualized by exposure to UV light and by exposure to an aqueous solution of ceric ammonium molibdate. The yield, mp, [α]^20^_D_, MS (ESI), ^1^H NMR, ^13^C NMR and IR of URB866 were previously reported [13].

### 2.3. Preparation of Polymeric Nanoparticles and Physicochemical Characterization of Nanoparticles

The PLGA nanoparticles were prepared according to the nanoprecipitation method of the pre-formed polymer in an aqueous solution [34]. Briefly, PLGA (12 mg, 0.6% *w*/*v*) was dissolved in acetone (2 mL) at room temperature, added to an aqueous phase (5 mL) containing PLX188 (1% *w*/*v*), homogenized at 24,000 rpm for 1 min (Ultraturrax T25, IKA^®^ Werke, Staufen, Germany) and then mechanically stirred at 600 rpm for 12 h in order to promote the evaporation of the organic solvent. Various amounts of URB866 (0.1–0.4 mg/mL) were added to the organic phase with the aim of favoring the nanoencapsulation of the molecule. The formulations were then purified by ultracentrifugation at 90 k × g for 60 min at 4 °C (Optima TL Ultracentrifuge, Beckman Coulter s.r.l., Milan, Itaky) or by means of Amicon^®^ Ultracentrifugal filters (cut-off 10 kDa, 4000 rpm for 20 min, Merck, Darmstadt, Germany) before the in vitro experiments.

The investigation of the mean diameter, the size distribution and the surface charge of the colloidal systems was performed by means of photon correlation spectroscopy (Zetasizer Nano ZS, Malvern Panalytical Ltd., Spectris plc, Malvern, UK) applying the third-order cumulant fitting correlation function, as previously described [34]. The results were expressed as the mean of three different experiments ± standard deviation. The stability profiles of the nanoparticles were investigated using Turbiscan Lab^®^ Expert (Formulaction, Toulouse, France) as a function of the temperature and incubation time. Specifically, the stability analysis of the nanoparticles was carried out evaluating the variation in the backscattering (ΔBS) and the transmittance (ΔT). Data were processed using a Turby Soft 2.0 and reported as the Turbiscan Stability Index (TSI) [33].

### 2.4. Evaluation of the Molecule Entrapment Efficiency and Release Profile

The amount of URB866 contained in the formulations was evaluated by means of spectrophotometric analyses. In detail, the polymeric suspension containing the molecule was centrifuged in order to separate the supernatant from the pellet, as previously described in Section 2.3. Successively, the pellet of PLGA nanosystems was dissolved in acetone, then the solvent was removed under a nitrogen flux and DMSO was added in order to solubilize only the active compound. After 6 h incubation, the solution was analyzed by a spectrophotometer (Lambda 35, Perkin Elmer, Waltham, MA, USA) at λ max 345. No interference deriving from the empty formulation was observed. The entrapment efficiency (EE%) was expressed as the following equation:EE% = De/Da × 100(1)
where De is the amount of entrapped URB866, and Da is the amount of compound added during the preparation of the systems. Moreover, the loading capacity (LC%) was described as the percentage of the total amount of entrapped molecule and the total weight of nanoparticles, according to the following equation:LC% = Amount of entrapped molecule/total weight of nanoparticles × 100(2)

The amount of PLX188 integrated within the colloidal structures was evaluated through colorimetric assays, exploiting the complex deriving from the interaction between the surfactant and the iodine solution, as previously reported [35].

The amount of URB866 released from the PLGA nanoparticles was evaluated through the dialysis method using cellulose acetate dialysis tubing (Spectra/Por with a molecular cut-off of 12,000–14,000 by Spectrum Laboratories Inc., Eindhoven, The Netherlands). A solution of PBS/ethanol (80:20) constantly stirred and warmed at 37 °C was used as the release fluid for the active compound. The nanoparticles containing URB866 (1 mL) were placed into dialysis bags (Spectrum Laboratories Inc., Eindhoven, The Netherlands), which were then transferred into beakers containing 200 mL of the release medium. At predetermined time intervals, 1 mL of it was withdrawn, removed under a nitrogen flux and resuspended in DMSO in order to analyze the compound as previously described. The release medium was replaced with a fresh one. The amount of released URB866 was calculated using the following equation:Release (%) = URB866_rel_/URB866_load_ × 100(3)
where URB866_rel_ is the amount of the released compound at time t, and URB866_load_ is the amount of the molecule entrapped within the colloidal particles.

### 2.5. Evaluation of URB866 Photostability

A xenon lamp irradiated the URB866 in the DMSO solution and the URB866-loaded nanosystems (molecule concentration equal to 240 μg/mL) in accordance with the ICH guideline for photostability testing (2021) (Photosafety evaluation of pharmaceuticals S10). Successively, the samples were analyzed using a UV–Vis spectrophotometer as a function of the exposure time, as previously described. The nanoparticles containing the bioactive were previously dissolved in DMSO (1 mL), while an empty formulation was used as the blank [36].

### 2.6. Cell Cultures

NCTC-2544 (human keratinocytes) and C-28 (human chondrocytes) cells were incubated in plastic culture dishes (100 mm × 20 mm) (Forma^®^ Series II Water-Jacketed CO_2_ Incubator (Thermo Fisher Scientific Milano srl, Milan, Italy) at 37 °C (5% CO_2_) using DMEM medium with glutamax, penicillin (100 UI/mL), streptomycin (100 μg/mL), amphotericin B (250 μg/mL) and FBS (10% *v*/*v*), as previously described [37]. Fresh medium was replaced every 48 h, and the cells were treated with trypsin when ∼80% confluence was reached, centrifuged at 1000 rpm at room temperature for 10 min, resuspended in an appropriate culture medium volume and seeded in culture dishes before in vitro investigation.

### 2.7. Investigation of the Antioxidant Activity

The antioxidant profile of URB866, both as a free molecule and encapsulated in polymeric nanoparticles, was assessed on both cell lines by means of MTT testing. In detail, the cells were plated in 96-well culture dishes (7 × 10^3^ cells/0.2 mL), incubated with different concentrations of URB866 (as a free form or nanoencapsulated within PLGA systems) for 24 h and then treated with hydrogen peroxide (800 μM) for 1.5 h. The effects of the treatment on the cell lines were assayed through MTT testing [33]. The antioxidant effects of URB866-loaded PLGA nanoparticles were normalized with respect to those exerted by the empty nanosystems at the same concentrations as the formulations containing the molecule.

### 2.8. Statistical Analysis

Statistical analysis of the various experiments was performed by ANOVA, and the results were confirmed by a Bonferroni *t*-test, with a *p* value of < 0.05 considered statistically significant.

## 3. Results and Discussion

### 3.1. Physicochemical and Technological Characterization

The physicochemical properties of PLGA nanoparticles prepared with PLX188 were investigated using photon correlation spectroscopy analysis, which demonstrated the formation of nanocarriers with a small diameter (120 nm) and a narrow size distribution (PI = 0.1). Indeed, the addition of the non-ionic surfactant provided a certain steric stability, preventing the aggregation of the particles. This was a consequence of the interaction between the hydrophobic residues of the poloxamer and PLGA [37,38]. A slight variation in the mean sizes (from ~120 nm to ~180 nm) was obtained when increased concentrations of URB866 (0.1 to 0.4 mg/mL) were added during the sample preparation, while the polydispersity index (PDI) remained constant, suggesting the absence of aggregates (Figure 2, Appendix A). These results were mostly influenced by the technique used, such as the nanoprecipitation method, as reported in several studies that showed a typical monomodal pattern of the obtained nanoparticles, characterized by a mean diameter of less than 160 nm and a PDI of less than 0.2 [39,40,41,42,43]. Moreover, the Z potential of the samples was constantly negative (~ −25 mV), demonstrating that the bioactive did not affect the surface charge of the colloidal systems. These features could be related to the lipophilic characteristic of the bioactive compound, which promoted its homogeneous localization in the polymer matrix, thus causing no significant variation in the colloidal properties [44,45]. Moreover, it was demonstrated that negatively charged nanocarriers are cleared more slowly from the blood than positively charged systems and are also non-genotoxic and non-cytotoxic [38,46].

The potential application of URB866-loaded polymeric nanocarriers as drug delivery systems requires the investigation of their physical stability as a function of the time and temperature by means of multiple light scattering. The evaluation of the backscattering and transmittance profiles of the nanoparticles containing the active compound demonstrated the great stability of the formulations at room temperature, evidencing no significant alteration of the aforementioned parameters with respect to the empty colloidal systems (Figure 3A–H). It is also interesting to observe that the increase in the temperature (from 25 °C to 37 °C) confirmed the absence of unfavorable physical phenomena such as sedimentation, flocculation or creaming, demonstrating the capacity of the nanocarriers to maintain stability over time at body temperature (Figure 3E–H).

In addition, the stability kinetic of the various formulations showed no significant variations in their TSI slopes at either 25 °C or 37 °C, with values of less than 3, suggesting that the colloidal systems are highly stable [47] (Figure 4).

### 3.2. Evaluation of the Retention Rate of URB866

The entrapment efficiency (EE) and the loading capacity (LC) are fundamental parameters to be evaluated during the pre-formulation of a drug delivery system because they influence the efficacious concentration of the active compound to be used and the in vivo therapeutic response. The investigation of the EE showed a proportional increase in the retention rate of the molecule when greater concentrations of URB866 were used during the preparation of the samples (Figure 5). In particular, the addition of 0.4 mg/mL of the molecule to the organic phase promoted a compound retention of ~60% (~0.24 mg/mL of the molecule) and a high loading capacity (~1.7). However, this value represents the maximum amount of the compound that can be used to obtain PLGA nanosystems containing URB866 because greater concentrations promoted the appearance of macroaggregates and sedimentation, probably because the loading saturation point was reached (data not shown). These findings are in agreement with several other experimental investigations showing that the higher the amount of molecule used, the greater the LC [44,48,49,50]. The in vitro release behavior of URB866 from PLGA nanoparticles is shown in Appendix A. As can be observed, the release profile of the molecule from the polymer matrix exhibited a biphasic pattern characterized by a fast initial release during the first 24 h, followed by a slower and prolonged release.

### 3.3. Photostability of URB866

The light sensitivity of URB866 is an issue that can compromise its application. For this reason, the capacity of PLGA nanoparticles to preserve the structure of the molecule was investigated by means of UV irradiation of the samples. A DMSO solution of the compound and the PLGA nanoparticles containing URB866 (molecule concentration of 240 µg/mL) were irradiated for different incubation times (Figure 6). The irradiation of the compound solution promoted a hypochromic effect of the absorbance, which was significantly reduced in the case of the molecule-loaded PLGA formulation. Indeed, the absorbance peak of URB866 after UV exposure shifted from 1.762 at 0 min to 1.080 after 240 min. On the contrary, the encapsulation of URB866 within polymeric nanoparticles elicited no significant variation in the main peak; in fact, the nanosystems promoted a lower decrease in absorbance following UV irradiation (from 1.722 at 0 min to 1.555 after 240 min) (Figure 6). These findings are related to the degradation of the molecule due to the phenomenon of photodegradation [14]. The results confirm the protection of URB866 contained within the polymeric matrix and the potential role of these systems as drug carriers able to minimize the negative effects exerted by the external stimuli on the encapsulated compound.

### 3.4. Evaluation of the Antioxidant Effects of URB866

The antioxidant effects of URB866 as a free compound or encapsulated in PLGA nanoparticles were investigated on two human cell lines, i.e., chondrocytes and keratinocytes, which could be potentially involved in oxidative stress reactions. In detail, the cells were pre-treated with the various formulations for 24 h and then incubated with hydrogen peroxide for 1.5 h in order to assess the ability of the various URB866 formulations to prevent the H_2_O_2_-induced cytotoxicity. As shown in Figure 7, the nanoencapsulation of the active compound promoted significant dose-dependent protection of cells with respect to the free molecule. URB866 exerted dose-dependent protection on stressed cells, and the entrapment of the compound in PLGA nanoparticles increased its antioxidant efficacy (cell viability of ~75% for NCTC-2544 and ~80% for C-28 with respect to ~45% for H_2_O_2_-treated cells in the absence of URB866 formulations). These results are consistent with several experimental investigations which have reported that the antioxidant activity of various compounds was preserved or increased after they were encapsulated within PLGA nanoparticles, whereas the empty nanosystems did not demonstrate any protective effect against oxidative stress [21,36,51,52,53,54,55]. These features suggest that the encapsulation of URB866 in PLGA nanoparticles may be a useful strategy for protecting the cells from oxidative stress thanks to the peculiar properties of the nanosystems (increased stability of the molecule and well-known cell uptake) able to better exploit the pharmacological effect of the molecule [45].

## 4. Conclusions

The short half-life of α-acylamino-β-lactone NAAA inhibitors is a noteworthy complication for the potential translation of the class molecules into oral pharmaceutical applications. As an example, therefore, the reference compound URB866 was encapsulated in PLGA nanoparticles to address this problem, obtaining a formulation able to preserve its physical stability and to demonstrate its potential as a new antioxidant drug. It is important to note that the use of polymer nanoparticles allowed the avoidance of this drawback due to the use of DMSO, an organic solvent which has been used to solubilize the molecule in the free form and associated with several adverse effects on humans [56]. The biocompatibility and biodegradability of the components of the formulation have already been approved for human applications, and this fact, as well as the intrinsic ability of the colloidal systems to protect the molecule from chemical, physical and enzymatic degradation, makes this formulation a suitable system to be used for the delivery of URB866 [57,58]. The results described herein proved to be interesting and preparatory for further investigations concerning the antioxidant and anti-inflammatory activity of PLGA nanoparticles loaded with α-acylamino-β-lactone NAAA inhibitors. Studies are ongoing to obtain further information on the efficacy and potential of this nanomedicine through in vitro and in vivo tests, in particular through specific models aimed at investigating the antioxidant properties of α-acylamino-β-lactone aromatic NAAA inhibitors.

## Figures and Tables

**Figure 1 antioxidants-11-00686-f001:**
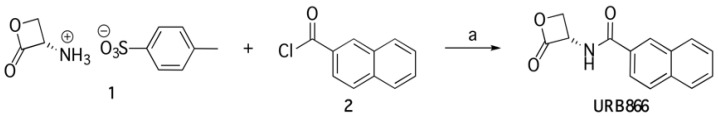
^a^ Reagents and conditions: (**a**) Et_3_N, CH_2_Cl_2_, 0 °C, 0.5 h, then room temperature, 2 h.

**Figure 2 antioxidants-11-00686-f002:**
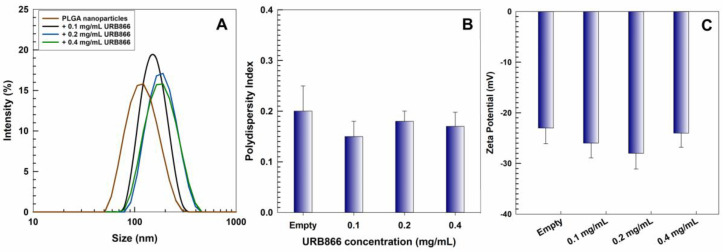
(**A**) Mean size, (**B**) polydispersity index and (**C**) zeta potential of PLGA nanoparticles as a function of the molecule concentration.

**Figure 3 antioxidants-11-00686-f003:**
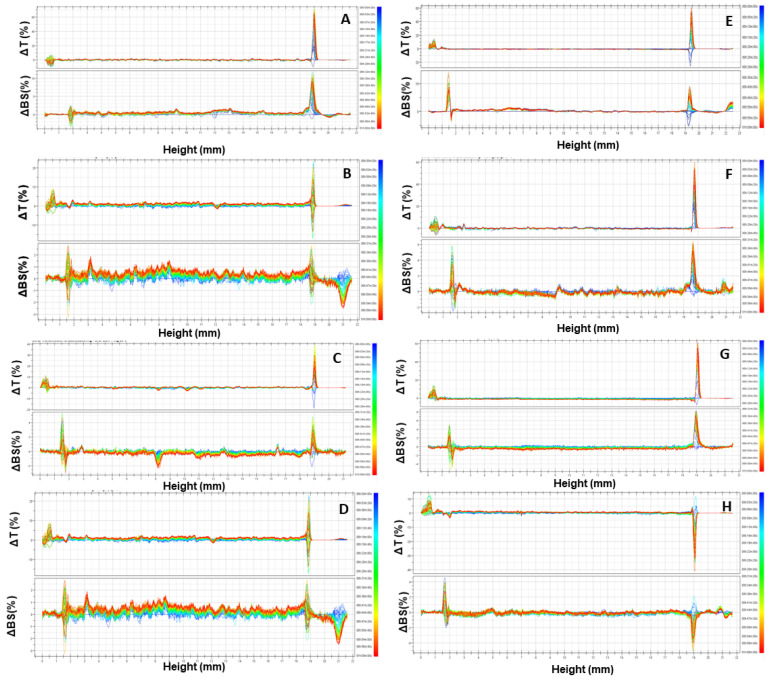
ΔT and ΔBS profiles of empty PLGA nanoparticles (**A**,**E**) and of nanosystems prepared with various amounts of URB866 [(**B**,**F**) 0.1 mg/mL; (**C**,**G**) 0.2 mg/mL; (**D**,**H**) 0.4 mg/mL] using Turbiscan Lab: (**A**–**D**) 25 °C; (**E**–**H**) 37 °C. The reported result is a representative experiment of three independent experiments.

**Figure 4 antioxidants-11-00686-f004:**
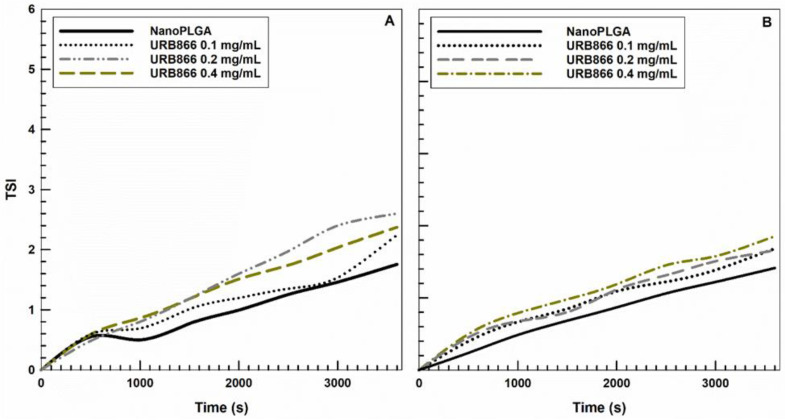
Turbiscan Stability Index (TSI) profile of PLGA nanoparticles containing various amounts of URB866: (**A**) 25 °C; (**B**) 37 °C.

**Figure 5 antioxidants-11-00686-f005:**
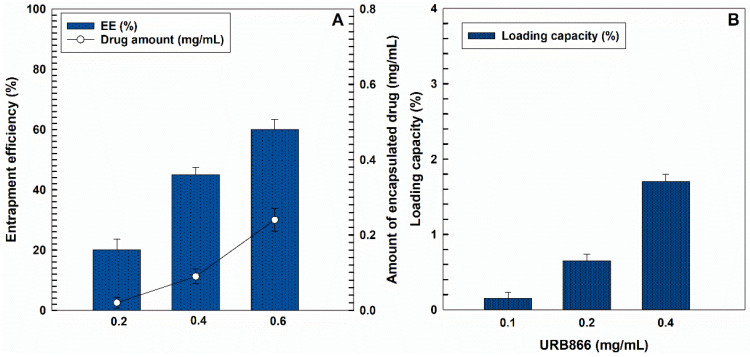
(**A**) Entrapment efficiency (EE, %) and (**B**) loading capacity (LC, %) of URB866 within PLGA nanoparticles as a function of the molecule concentration used during the sample preparation.

**Figure 6 antioxidants-11-00686-f006:**
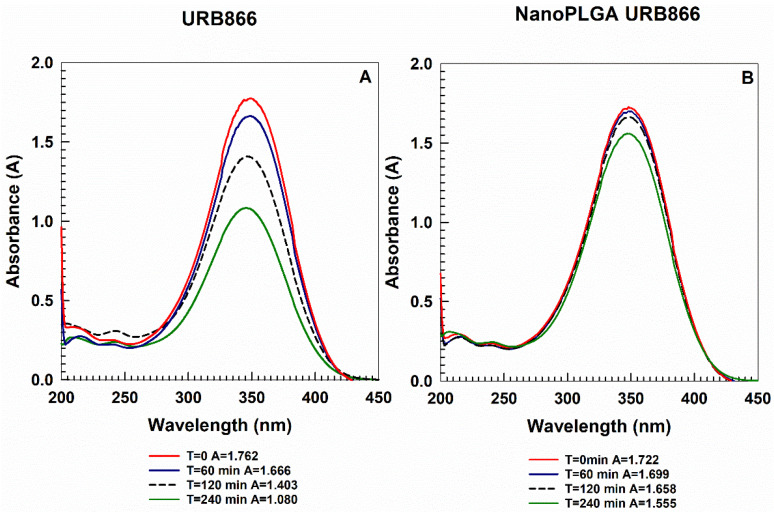
Photostability of URB866 (molecule concentration of 240 μg/mL) as (**A**) a DMSO solution or (**B**) entrapped in PLGA nanoparticles after UV irradiation as a function of the incubation time.

**Figure 7 antioxidants-11-00686-f007:**
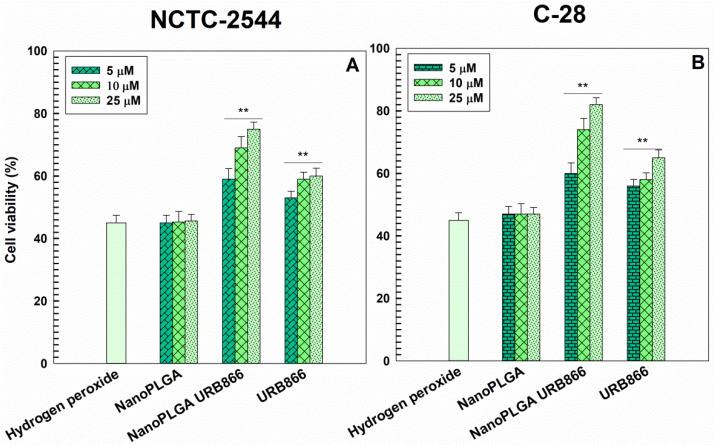
Antioxidant activity of URB866 in the free form or encapsulated within PLGA nanoparticles on NCTC2544 (**A**) and C-28 cells (**B**). The cells were incubated with different concentrations of URB866 for 24 h and then treated with H_2_O_2_ (800 µM) for 1.5 h. At the end of the incubation time, the MTT test was carried out. The results are the mean of three different experiments ± standard deviation. Statistical analysis: **, *p* <0.001 vs. hydrogen peroxide.

## Data Availability

All data available are reported in the article.

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
