# Peer review of "α-Acylamino-β-lactone N-Acylethanolamine-hydrolyzing Acid Amidase Inhibitors Encapsulated in PLGA Nanoparticles: Improvement of the Physical Stability and Protection of Human Cells from Hydrogen Peroxide-Induced Oxidative Stress"

_antioxidants, 2022, doi:10.3390/antiox11040686_

Round 1

Reviewer 1 Report

Very nice work, well organised and clearly described. The paper can be accepted after correcting minor points:

  • Please, complete the caption from Figure 3 as E-H plots were not identified.
  • Ln 274: are in agreement
  • Figures 5-7: Use the letters A, B to identify left-hand and right-hand plots, respectively, and describe them in each figure caption.

Author Response

The authors are very grateful to the Reviewer for the valued queries and advices. It is the opinion of the authors that the following changes in the manuscript have improved the quality of the paper. A response to each point raised in the main text has been shown in red.

Reviewer 1 (R1): Very nice work, well organised and clearly described. The paper can be accepted after correcting minor points:

R1: Please, complete the caption from Figure 3 as E-H plots were not identified.

Authors (A): The authors are very grateful to the Reviewer for such a positive evaluation. The plots in the figures have been duly reported in the caption.

R1: Ln 274: are in agreement

A: The sentence was duly modified as suggested by the Reviewer (line 312).

R1: Figures 5-7: Use the letters A, B to identify left-hand and right-hand plots, respectively, and describe them in each figure caption.

A: In response to the Reviewer’s request, the letters have been duly added in the figures and described in their respective captions.

Reviewer 2 Report

The studies presented for evaluation are a continuation of the research on the instance molecule, but not a drug, which the Authors should bear in mind. Therefore, the word "drug" should not appear in the work, only the molecule shows biological / pharmacological activity (depending on whether preclinical tests were carried out).

The work requires numerous changes before publication. The results of obtaining the system have been well presented, whatever its characteristics require supplementing. The authors should describe what changes the molecule undergoes after its introduction into systems (e.g. solubility studies, in vitro permeation).

The work describes only the results of the photostability tests. Photostability is an example of chemical stability. Physical stability includes research on changing the crystalline form of active compounds or obtaining amorphous dispersions. The assessment of the photostability should be performed using the HPLC technique. The degradation products  generated during decomposition may have similar UV spectra, therefore non-selective techniques are not suitable for such studies.

The authors should also broadly assess the chemical stability of the compound after its introduction into the system (e.g. the influence of temperature or humidity).

Assessment of the effect of hydrogen peroxide on the stability of molecules is also a stage of stress tests, other test models (eg DPPH, Frap techniques) are used to assess the antioxidant potential.

I believe that the Authors should significantly organize the presentation of the results by referring to the ICH guidelines for research on innovative molecules.

Author Response

The authors are very grateful to the Reviewer for the valued queries and advices. It is the opinion of the authors that the following changes in the manuscript have improved the quality of the paper. A response to each point raised the main text has been shown in red.

Reviewer 2 (R2): The studies presented for evaluation are a continuation of the research on the instance molecule, but not a drug, which the Authors should bear in mind. Therefore, the word "drug" should not appear in the work, only the molecule shows biological / pharmacological activity (depending on whether preclinical tests were carried out).

Authors (A): According to the Reviewer’s suggestion, the word "drug" has been duly modified in the main text.

R2: The work requires numerous changes before publication. The results of obtaining the system have been well presented, whatever its characteristics require supplementing. The authors should describe what changes the molecule undergoes after its introduction into systems (e.g. solubility studies, in vitro permeation).

The authors should also broadly assess the chemical stability of the compound after its introduction into the system (e.g. the influence of temperature or humidity).

A: The authors thank the Reviewer for the interesting question. Nevertheless, the molecule URB866 has been already characterized from a chemical point of view as reported in a previous published manuscript (Duranti et al., 2012, Journal of medicinal chemistry, 2012, 55: 4824-4836, https://doi.org/10.1021/jm300349j) and a comment on this is now reported in the text (lines 74-77). It is the opinion of the authors that a brief description of the basic concepts governing the nanoencapsulation would be useful towards providing a conceivable answer to the Reviewer’s concern. The nanoencapsulation involves the physico-chemical retention of an active compound within colloidal or molecular carriers. This approach usually preserves the entrapped molecules from adverse environmental conditions and undesirable effects exerted by light, oxygen, pH, moisture, heat, shear, or other extreme conditions, by means of a protective barrier surrounding the compound. The nanoencapsulation favors the masking of undesirable flavors, the reduction of evaporation of volatile substances and the increase of physical stability, biological activity and shelf-life of active compounds (Shishir et al., 2018, Trends in Food Science & Technology, 78: 34-60, https://doi.org/10.1016/j.tifs.2018.05.018; Saka and Chella, 2020, Environmental Chemistry Letters, 19: 1097-1106, https://doi.org/10.1007/s10311-020-01103-9; Ngwuluka et al., 2020; Zhu, 2017, Food Chemistry, 229:542-552, https://doi.org/10.1016/j.foodchem.2017.02.101; Fathi et al., 2014, Trends in Food Science & Technology, 39:18-39, https://doi.org/10.1016/j.tifs.2014.06.007).

The aim of the proposed investigation is i) to characterize PLGA nanosystems containing URB866, as a representative tool of the class of NAAA inhibitors considered, demonstrating that the nanoencapsulation of the compound can ii) preserve its integrity from external stresses (for example adverse phenomena related to light exposition) and iii) its pharmacological effects. It is opinion of the authors that the experiment of photostability demonstrates that the polymeric matrix is able to maintain the structural integrity of the entrapped compound with respect to the free form of the molecule.

The authors agree with the Reviewer’s statement about the opportunity to investigate the specific interactions occurring between URB866, PLGA and poloxamer 188 and these experiments will be performed in order to better understand the phenomena promoting the retention and release of the compound but these specific aspects were not evaluated in the proposed topic as this is a preliminary study.

Moreover, the in vitro permeation of the compound through a biological membrane requires a study of intracellular accumulation or the use of a radiolabelled derivative and it is an investigation out of the proposed topic; on the other hand, the fact that PLGA nanoparticles can be efficiently uptaken by the cells, increasing the cytosolic localization of the entrapped compound is a well-known concept already demonstrated by several research teams in the last decade.

Finally, the nanoencapsulation of a lipophilic molecule (such as URB866) in a colloidal polymeric system allows the use of polar media for its administration thanks to the interface phenomena occurring between the particles and the solvent, avoiding the use of organic compounds, such as DMSO, and the related potential side effects (lines 384-387).

R2: The work describes only the results of the photostability tests. Photostability is an example of chemical stability. Physical stability includes research on changing the crystalline form of active compounds or obtaining amorphous dispersions. The assessment of the photostability should be performed using the HPLC technique. The degradation products generated during decomposition may have similar UV spectra, therefore non-selective techniques are not suitable for such studies.

A: The authors agree with the statement of the Reviewer. Indeed, there are many methods to investigate the chemical stability of a novel compound, such as X-ray diffraction (XRD), after its encapsulation in a colloidal system. However, as interesting as it may be, this investigation is not the aim of the proposed manuscript. Indeed, the purpose of the current experimental work was to demonstrate the ability of PLGA nanosystems to prevent the rapid degradation of URB866, increasing its physical stability after exposure to light with respect to the free form of the molecule.

Moreover, even though the HPLC can provide additional information about the products of degradation of URB866, the idea of the authors is to demonstrate that the compound can be efficiently protected when it is nanoencapsulated. For this purpose, according to the I.C.H. guidelines (The ICH Q1B test), the UV-visible method was very appropriated, as demonstrated in several other experimental studies (Swetledge et al., 2021, Scientific reports, 11:12270, https://doi.org/10.1038/s41598-021-90792-5; Pegoraro et al., 2018, Pharmaceutical development and technology, 23:400-406, https://doi.org/10.1080/10837450.2017.1332641; Cosco et al., 2011, Journal of drug delivery science and technology, 21:395-400, https://doi.org/10.1016/S1773-2247(11)50064-4; Chopra et al., 2016, Biomaterials, 2016, 84: 25-41, https://doi.org/10.1016/j.biomaterials.2016.01.018; Li et al., Food Hydrocolloids, 2019, 87: 342-351. https://doi.org/10.1016/j.foodhyd.2018.08.002; Sorasitthiyanukarn et al., 2018, Materials Science and Engineering: C, 93:178-190. https://doi.org/10.1016/j.msec.2018.07.069).

R2: Assessment of the effect of hydrogen peroxide on the stability of molecules is also a stage of stress tests, other test models (eg DPPH, Frap techniques) are used to assess the antioxidant potential.

A: Authors agree with the Reviewer’s statement. However, the findings discussed in the proposed manuscript are an evidence of the protective properties against oxidation exerted by URB866 when it is encapsulated within PLGA nanoparticles, and the obtained results need additional investigations in order better understand the involved biological and biochemical phenomena. Indeed, as reported in the Conclusions section (lines 373-379) “The results herein described proved to be interesting and preparatory for further investigations concerning the antioxidant and anti-inflammatory activity of PLGA nanoparticles loaded with α-acylamino-β-lactone NAAA inhibitors. Studies are ongoing to obtain further information on the efficacy and potential of this nanomedicine through in vitro and in vivo tests, in particular through specific models aimed at investigating the antioxidant properties of α-acylamino-β-lactone aromatic NAAA inhibitors”.

R2: I believe that the Authors should significantly organize the presentation of the results by referring to the ICH guidelines for research on innovative molecules.

A: The ICH (International Conference on Harmonization) guidelines describe the stability tests for novel compounds over time as a function of different environmental storage conditions (pH, temperature, light, air, and humidity). Nevertheless, the purpose of this study was not to characterize the molecule URB866, on which in-depth studies were already done in a previous work (Duranti et al., 2012, Journal of Medicinal Chemistry, 2012, 55:4824-4836, https://doi.org/10.1021/jm300349j), but to address the criticisms concerning the short half-life of α-acylamino-β-lactone NAAA inhibitors by their encapsulation within biocompatible polymeric nanoparticles, preserving their pharmacological properties.

Reviewer 3 Report

Present manuscript discuss the synthesis and characterization of PLGA nanoparticles for delivery of N-acylethanolamine acid amidase (NAAA) inhibitor called URB866.

1) Synthesis of PLGA Nanoparticles: section 2.3. How did you do evaporation of solvent (i am assuming that you are doing acetone evaporation) was it at room temperature or vacuum?

And how did you make sure that solvent was completely evaporated, unbound drug was discarded?

What is the role of PLX188 in synthesis is it working as stabilizer? 

2) Section 2.4: when drug was solubilized in acetone why to dry it and reconstitute in DMSO for UV reading?

3) It is unclear why FTIR or NMR analysis to investigate drug-plga nanoparticle was not performed when they have data for standard drug. This will help to check if the drug is nanoconjugate or classical self assembly based encapsulation.

4) If possible it will be good to have drug release kinetic data.

5) Can you justify why do you think that antioxidant effect for drug-PLGA group in figure 7 is relatively higher than native drug? 

Round 2

Reviewer 3 Report

I think authors have answered all the major concerns and it can be accepted for publication.